# Angiogenic Activity of Cytochalasin B-Induced Membrane Vesicles of Human Mesenchymal Stem Cells

**DOI:** 10.3390/cells9010095

**Published:** 2019-12-30

**Authors:** Marina O. Gomzikova, Margarita N. Zhuravleva, Vyacheslav V. Vorobev, Ilnur I. Salafutdinov, Alexander V. Laikov, Sevindzh K. Kletukhina, Ekaterina V. Martynova, Leysan G. Tazetdinova, Atara I. Ntekim, Svetlana F. Khaiboullina, Albert A. Rizvanov

**Affiliations:** 1Openlab “Gene and cell technologies”, Institute of Fundamental Medicine and Biology, Kazan Federal University, 420008 Kazan, Russia; k.i.t.t.1807@gmail.com (M.N.Z.); sal.ilnur@gmail.com (I.I.S.); sevindzh.rasulova.1993@mail.ru (S.K.K.); ignietferro.venivedivici@gmail.com (E.V.M.); safinaleys@gmail.com (L.G.T.); sv.khaiboullina@gmail.com (S.F.K.); 2M.M. Shemyakin–Yu.A. Ovchinnikov Institute of Bioorganic Chemistry of the Russian Academy of Sciences, 117997 Moscow, Russia; 3Faculty of Medicine and Health Sciences, University of Nottingham, Nottingham LE12 5RD, UK; 4Department of Radiation Oncology, College of Medicine, University of Ibadan, Ibadan 200284, Nigeria; 5Department of Microbiology and Immunology, Reno School of Medicine, University of Nevada, Reno, NV 89557, USA

**Keywords:** extracellular vesicles, microvesicles, membrane vesicles, cytochalasin B, mesenchymal stem cells, angiogenesis, cell-free therapy

## Abstract

The cytochalasin B-induced membrane vesicles (CIMVs) are suggested to be used as a vehicle for the delivery of therapeutics. However, the angiogenic activity and therapeutic potential of human mesenchymal stem/stromal cells (MSCs) derived CIMVs (CIMVs-MSCs) remains unknown. Objectives: The objectives of this study were to analyze the morphology, size distribution, molecular composition, and angiogenic properties of CIMVs-MSCs. Methods: The morphology of CIMVs-MSC was analyzed by scanning electron microscopy. The proteomic analysis, multiplex analysis, and immunostaining were used to characterize the molecular composition of the CIMVs-MSCs. The transfer of surface proteins from a donor to a recipient cell mediated by CIMVs-MSCs was demonstrated using immunostaining and confocal microscopy. The angiogenic potential of CIMVs-MSCs was evaluated using an in vivo approach of subcutaneous implantation of CIMVs-MSCs in mixture with Matrigel matrix. Results: Human CIMVs-MSCs retain parental MSCs content, such as growth factors, cytokines, and chemokines: EGF, FGF-2, Eotaxin, TGF-α, G-CSF, Flt-3L, GM-CSF, Fractalkine, IFNα2, IFN-γ, GRO, IL-10, MCP-3, IL-12p40, MDC, IL-12p70, IL-15, sCD40L, IL-17A, IL-1RA, IL-1a, IL-9, IL-1b, IL-2, IL-4, IL-5, IL-6, IL-7, IL-8, IP-10, MCP-1, MIP_1a, MIP-1b, TNF-α, TNF-β, VEGF. CIMVs-MSCs also have the expression of surface receptors similar to those in parental human MSCs (CD90^+^, CD29^+^, CD44^+^, CD73^+^). Additionally, CIMVs-MSCs could transfer membrane receptors to the surfaces of target cells in vitro. Finally, CIMVs-MSCs can induce angiogenesis in vivo after subcutaneous injection into adult rats. Conclusions: Human CIMVs-MSCs have similar content, immunophenotype, and angiogenic activity to those of the parental MSCs. Therefore, we believe that human CIMVs-MSCs could be used for cell free therapy of degenerative diseases.

## 1. Introduction

Biological activity and differentiation potential of the mesenchymal stem/stromal cells (MSCs) made them an attractive tool for regenerative medicine [1]. However, the risk of undesirable effects, including malignant transformation [2] and differentiation in unwanted directions [3,4], limits their therapeutic application.

Recent discoveries have suggested a key role of the intercellular communication in the MSCs effects on the neighboring cells. Important mediators of intercellular communication are extracellular vesicles (EVs)-membrane-enclosed nano- and microstructures released by mammalian cells [5]. It was shown that MSCs-derived EVs contain biologically active molecules, similar to those in the donor cells. In fact, several studies showed that EVs retain the therapeutic effect of the donor MSCs [6]. For example, the regenerative potential of MSCs-derived EVs was demonstrated using the rat model of myocardial infarction, where EVs restored the blood flow and reduced the size of the tissue damage [7]. In the mouse model of hindlimb ischemia EVs induced formation of new blood vessels and increased blood reperfusion [8]. MSC-EVs mediate cartilage repair by enhancing proliferation of chondrocytes, attenuating apoptosis, and modulating immune reactivity [9]. MSC-EVs have also been reported to reduce the hepatic injury by improving the hepatocytes viability, suppressing oxidative injury and modulating the inflammatory response in vivo [10]. MSCs-derived EVs were also successfully applied to treat transient global ischemia in mice, where the neuroprotective effects of MSC-EVs were demonstrated [11]. Based on these findings, the cell-free therapy concept was developed, where the MSCs-derived EVs are used instead of cells [12,13,14].

EVs represent a heterogeneous population of vesicles including microparticles (or microvesicles), exosomes, and apoptotic bodies [15]. Microvesicles are microstructures budding from the cell surface that range from 0.1 to 1 μm in diameter [15]. Exosomes are nanostructures of endogenous origin 40–100 nm in size [15]. Apoptotic bodies are large microstructures more than 1 μm in diameter, which are released from dying cells and mostly contain fragmented DNA [16].

EVs can be used as a therapeutic vehicle as they have multiple advantages: (1) Microvesicles retain the surface proteins of parental cells, (2) hydrophobic molecules (bioactive lipids and membrane proteins) could be transferred by EVs, and (3) the EVs cytoplasmic membrane protects their content from degradation [12]. However, the yield of the naturally produced EVs by MSCs is low, limiting their clinical application. One of the strategies to enhance the EVs’ yield is using bioreactor cultures of bone marrow MSCs [17]. Furthermore, the osmotic shock was applied to induce the release of vesicles from CHO cells [18]. In another approach, the cells suspension extrusion through the polycarbonate filter with 1, 2, or 3 μm-pore size was applied to isolate the plasma membrane vesicles [19]. A more moderate large scale EVs production was achieved by using cytochalasin B treatment of target cells [20].

The potential of cytochalasin B-induced membrane vesicles (CIMVs) as a vector for drug delivery has been demonstrated [21]. The main question concerning the production of cytochalasin B-induced membrane vesicles was how different they were from apoptotic bodies since there was a concern that treatment of cells with a drug may induce apoptosis. Peng et al. demonstrated that cytochalasin B does not cause cell death, membrane vesicles possess surface charge similar to their parental cells, and concluded that cytochalasin B-induced membrane vesicles are not apoptotic bodies [21]. Moreover, we have previously found that neither cytochalasin B treatment of donor cells, nor treatment of recipient cells with CIMVs induce cell death [22]. CIMVs have a size similar to that of natural EVs [22]. CIMVs also retain the biological activity of parental cells and are able to stimulate capillary tube formation in vitro and vasculogenesis in vivo [22]. These properties of CIMVs led to our interest in the evaluation of the biological activity of MSCs-derived CIMVs (CIMVs-MSCs) as this might provide a potential tool for cell-free therapy.

The objectives of this study were to determine the morphology, molecular content, and angiogenic activity of human CIMVs-MSCs, as well as describe the membrane receptor transferring and biological activity of human CIMVs-MSCs.

## 2. Materials and Methods

### 2.1. MSCs Isolation and Characterization

Human samples were collected and methods were carried out in accordance with an experimental protocol approved by the Biomedicine Ethic Expert Committee of Kazan Federal University and Republican clinical hospital (No. 218, 11.15.2012) based on article 20 of the Federal Legislation on “Health Protection of Citizens of the Russian Federation” № 323-FL, 21 November 2011. Signed informed consent was obtained from all donors. To obtain cell suspension, the adipose tissue was cut into small pieces of about 1–3 mm^3^ in size, then 10 mL of minced fat were transferred into a 50 mL tube. Next, an equal volume of 0.2% collagenase II solution (Dia-M, Russia) was added and the mixture was incubated in a shaker-incubator at 37 °C, 120 rpm for one hour. After centrifugation (400× *g* for 5 min), the upper fat layer was discarded, the supernatant was removed, and the remaining cell pellet was washed once in PBS (PanEco, Moscow, Russia). Then cells were re-suspended in DMEM (PanEco, Moscow, Russia) supplemented with 10% fetal bovine serum (Gibco, UK) and 2 mM L-glutamine (PanEco, Moscow, Russia). To remove the remaining tissue parts, the suspension was filtered through a cell strainer (40 µm, 93040, SPL, Korea) into a fresh tube. The cell suspension was transferred into a culture flask (ratio for solid adipose tissue was 175 cm^2^ surface area/10–15 mL of adipose tissue). The culture medium was changed after 1 day of culture and the cells were maintained in a humidified environment at 37 °C, 5% CO_2_ with culture medium replaced every three days. 

Adipose tissue-derived MSCs were differentiated into the three lineages: adipogenic, chondrogenic, and osteogenic. The StemPro™ Adipogenesis Differentiation Kit (A1007001, ThermoFisher Scientific, Waltham, MA, USA), the StemPro™ Chondrogenesis Differentiation Kit (A1007101, ThermoFisher Scientific, Waltham, MA, USA), and the StemPro™ Osteogenesis Differentiation Kit (A1007201, ThermoFisher Scientific, Waltham, MA, USA) were used to induce the differentiation in accordance with the manufacturer’s instructions. Briefly, MSCs were seeded at 1 × 10^4^ cells/cm^2^ (for adipogenic differentiation) or 5 × 10^3^ cells/cm^2^ (for osteogenic differentiation). For chondrogenic differentiation, a cell suspension (1.6 × 10^7^ cells/mL) was made to generate micromass culture, complete differentiation medium was replaced every three days. Twenty-one days later the adipogenic, chondrogenic, and osteogenic differentiation was confirmed by detection of lipid droplets (Oil Red dye staining), glycosaminoglycans and mucins (1% alcian blue staining), and calcium deposits (5% AgNO_3_ staining), respectively [23].

The immune phenotype of isolated cells was analyzed by staining with monoclonal antibodies CD90-PE/Cy5 (328112; BioLegend, San Diego, CA, USA), CD90-Brilliant Violet 421 (328122; BioLegend, San Diego, CA, USA); CD44-APC/Cy7 (103028; BioLegend, San Diego, CA, USA), CD29-APC (2115040; Sony, San Jose, CA, USA), CD73-APC (51-9007649; BD bioscience, San Jose, CA, USA), CD73-PerCP-Cy5.5 (344014; BioLegend, San Diego, CA, USA), STRO-1-APC/Cy7 (340104; BioLegend, San Diego, CA, USA), CD45-FITC (304006; BioLegend, San Diego, CA, USA). Expression of CD markers were analyzed by flow cytometry using BD FACS Aria III (BD bioscience, San Jose, CA, USA).

### 2.2. CIMVs Production

CIMVs were prepared as described previously [22]. Briefly, MSCs of passage 4 were used in the study. After reaching a confluence of 80–90%, the MSCs were detached using trypsin/EDTA solution (2 mL/T75 flask). After 5 min incubation at 37 °C, 5% CO_2_, trypsin was inactivated by adding the culture medium. MSCs were washed twice with PBS and maintained in DMEM supplemented with 10 µg/mL of cytochalasin B (Sigma-Aldrich, St. Louis, MO, USA) for 30 min (37 °C, 5% CO_2_). Cell suspension was vortexed vigorously for 30 sec and pelleted (100× *g* for 10 min). The supernatant was collected and subject to two subsequent centrifugation steps (100× *g* for 20 min and 2000× *g* for 25 min). The pellet from the last step, containing CIMVs-MSC, was washed once in PBS (2000× *g* for 25 min).

### 2.3. Characterization of the CIMVs

#### 2.3.1. Scanning Electron Microscopy (SEM)

CIMVs were fixed (10% formalin for 15 min) and dehydrated using graded alcohol series and dried at 37 °C. Prior to imaging, samples were coated with gold/palladium in a Quorum T150ES sputter coater (Quorum Technologies Ltd., Lewes, United Kingdom). Slides were analyzed using Merlin field emission scanning electron microscope (CarlZeiss, Oberkochen, Germany). For the size analysis, three independent batches of CIMVs-MSCs (MSCs were obtained from three donors) were produced and used to generate at least six electron microscope images for each batch. Data collected was used to determine the CIMVs size.

#### 2.3.2. Proteome Analysis

CIMVs derived from 3 × 10^6^ MSCs and 1 × 10^6^ MSCs were lysed in RIPA buffer (150 mM NaCl, 1% NP-40, 0.5% sodium deoxycholate, 0.1% SDS, 25 mM Tris (pH 7.4)) and separated using gel polyacrylamide gel electrophoresis (20 μg of total protein per well) [24]. Gels were fixed overnight (20% ethanol and 10% acetic acid), strips were cut (1.5 × 1.5 mm), dehydrated using 100% acetonitrile for 20 min, and dried at room temperature. Gel fragments were rehydrated (200 mM ammonium bicarbonate, 100% acetonitrile, dH2O), placed in sequencing grade-modified trypsin (working concentration was 20 ng/μL) (Promega, Madison, WI, USA), and incubated overnight at 37 °C. The cleaved peptides were extracted from the gel pieces using extraction buffer (0.5% trifluoroacetic acid (TFA)) and incubated in an ultrasonic bath for 10 min, followed by adding 100% acetonitrile and 0.5% TFA. The mixture of peptides was dried at 45 °C under vacuum using Concentrator plus Complete System (Eppendorf, Hamburg, Germany). Samples were desalted using the Acclaim PepMap 100 Columns (160321, Thermo Scientific, Waltham, MA, USA) (C18, 3 µm, 100 Å) for 5 min at a 5 µL/min flow rate.

Liquid chromatography mass spectrometry analysis (LC-MS/MS) of peptide extracts was done using the 3000 Nano LC nanochromatographic system (Thermo Scientific, Waltham, MA, USA) and Maxis Impact mass spectrometer with an electrospray ionization source Captive Spray (Bruker, Billerica, MA, USA). Peptides were separated by reverse phase chromatography using an Acclaim PepMap 100 NanoViper column (C18, 2 µm, 100 Å, 75 µm × 15 cm) (Thermo Scientific, Waltham, MA, USA). The tryptic peptides were eluted in a linear gradient with a mixture of solution A (4.5% acetonetrile in diH2O with 0.5% formic acid) and increasing percentage (from 5 to 35%) of solution B (94.5% acetonetrile with 0.5% formic acid). Elution was done for 60 min at 40 °C and a 300 nL/min flow rate, 1600 V of the Captive Spray source, capillary temperature 150 °C, and 3.0 L/min dry gas flow. The positive polarity and total spectrum measurements, as well as data dependent acquisition (DDA) were set on Maxis Impact mass spectrometer (Bruker, USA). Peptides mass spectrum was compared to the theoretical peptide masses of all human proteins using the SWISS-PROT and NCBI databases.

#### 2.3.3. Multiplex Analysis

Multiplex analysis based on the xMAP Luminex technology was performed with the use of MILLIPLEX MAP Human Cytokine/Chemokine Magnetic Bead Panel—Premixed 38 Plex-Immunology Multiplex Assay (sCD40L, EGF, Eotaxin/CCL11, FGF-2, Flt-3 ligand, Fractalkine, G-CSF, GM-CSF, GRO, IFN-α2, IFN-γ, IL-1α, IL-1β, IL-1ra, IL-2, IL-3, IL-4, IL-5, IL-6, IL-7, IL-8, IL-9, IL-10, IL-12 (p40), IL-12 (p70), IL-13, IL-15, IL-17A, IP-10, MCP-1, MCP-3, MDC/CCL22, MIP-1α, MIP-1β, TGF-α, TNF-α, TNF-β, VEGF) (Merckmillipore, Burlington, MA, USA), in accordance with the manufacturer’s instructions. Briefly, samples were incubated with fluorescent beads for 1 h, washed and incubated with phycoerythrin-streptavidin for 10 min (Merckmillipore, Burlington, MA, USA). The analysis was done using a Luminex 200 analyzer (Merckmillipore, Burlington, MA, USA). The CIMVs-MSCs and MSCs lysates in IP buffer (50 mMTris-Cl, 150 mMNaCl, 1% Nonidet-P40) were used for multiplex analysis. Equal protein load (25 μg) was used for the analysis.

#### 2.3.4. Flow Cytometry Analysis

The immune phenotype of CIMVs-MSCs was analyzed by immunostaining with monoclonal antibodies: CD90-PE-Cy5 (328112; BioLegend, San Diego, CA, USA), CD29-APC (2115040; Sony, San Jose, CA, USA), CD44-APC/Cy7 (103028; BioLegend, San Diego, CA, USA), CD73-PerCP/Cy5.5 (344014; BioLegend, San Diego, CA, USA). CIMVs were analyzed by flow cytometry (BD FACS Aria III, BD Bioscience, San Jose, CA, USA), and the 405 nm laser was used for better a resolution of CIMVs-MSC. 

#### 2.3.5. Cytoplasmic Membrane Staining

CIMVs were stained with lipophilic dye DiD (V22889; Life Technologies, Carlsbad, CA, USA) according to the manufacturer′s instructions. Briefly, CIMVs (300 μg/mL) were incubated in 5 µM of DiD dye for 15 min (37 °C, 5% CO_2_) and washed twice with complete medium (DMEM with 10% FBS, 2 mM l-glutamine) before use.

### 2.4. Animals

Adult rats (Rattus norvegicus) (Pushchino, Russia) were used. All experiments were carried out in compliance with the procedure protocols approved by the Kazan Federal University (KFU) local ethics committee (protocol #5, date: 27 May 2014) according to the rules adopted by KFU and Russian Federation Laws. All experiments were repeated three times. Rats were euthanized using CO_2_ in compliance with the procedure protocols approved by the KFU local ethics committee (protocol #5, date: 27 May 2014). Immunocompetent animals were used in the experiment due to the fact that MSCs have low immunogenicity and immunosuppressive effect [25], as well as the short duration of the experiment (specific immunity on the introduced cells was not developed) [26].

### 2.5. Angiogenic Activity Test In Vivo

Rats were injected with MSCs (1 × 10^6^) or CIMVs (50 µg) in 400 µL Matrigel Basement Membrane Matrix (10 mg/mL) (356231, BectonDickinson, USA) subcutaneously. Animals were injected with MSCs and CIMVs-MSCs unstained or pre-stained with DiO membrane dye (V-22886, LifeTechnoligies, Carlsbad, CA, USA). Control animals were injected with the Matrigel matrix (10 mg/mL). Each experimental group included six animals: (1) Matrigel matrix injection; (2) MSCs (10^6^ cells) in Matrigel Matrix; (3) CIMVs-MSCs (50 µg) in Matrigel Matrix; (4) pre-stained with DiO MSCs (10^6^ cells) in Matrigel Matrix; (5) pre-stained with DiO CIMVs-MSCs (50 µg) in Matrigel Matrix. Eight days later, fragments of Matrigel matrix were collected, fixed with 10% buffered formalin solution (BioVitrum, Saint-Petersburg, Russia), dehydrated in an ethanol gradient, and embedded into Histomix paraffin (BioVitrum, Saint-Petersburg, Russia). Paraffin sections (6 μm thick) were cut using HM 355S microtome (ThermoScientific, Waltham, MA, USA), dewaxed with Roticlear (CarlRoth, Karlsruhe, Germany), and stained with hematoxylin-eosin (BioVitrum, Saint-Petersburg, Russia). Then the sections were dehydrated and embedded in a Canadian balsam (PanReac AppliChem, Chicago, IL, USA). Slides were examined using the AxioOberver.Z1 (CarlZeiss, Oberkochen, Germany) fluorescent microscope. Five fragments of Matrigel matrix and at least three microscope images were examined for each batch.

Fragments of Matrigel matrix containing MSCs or CIMVs pre-stained with DiO dye (V-22886, LifeTechnoligies, Carlsbad, CA, USA) were stored in liquid nitrogen. Matrigel matrix slides (6 μm thick) were made using HM560 Cryo-Star microtome (ThermoScientific, Waltham, MA, USA). Nucleus was stained using 5 μg/mL DAPI (D1306, Invitrogen, USA).

### 2.6. Statistical Analysis

Statistical analysis was done using Wilcoxon signed-rank test (R-Studio) with significance level *p* < 0.05. Illustrations were built with the “ggplot2” package (v3.1.0, 2018).

## 3. Results

### 3.1. Isolation and Characterization of Human Adipose-Derived MSCs

Primary MSCs were isolated from human subcutaneous adipose tissue. MSCs phenotype was confirmed using antibodies against CD90, CD73, CD44, CD105, and CD45 (Figure 1A). Next, differentiation potential (chondrogenic, adipogenic, and osteogenic) of isolated MSCs were analyzed (Figure 1B).

Cells phenotype were determined as CD90^+^, CD44^+^, CD29^+^, CD73^+^, STRO-1^+^, and CD45^−^ (Figure 1A), which are characteristics of the MSCs [27]. Isolated cells could be differentiated into chondrogenic, adipogenic, and osteogenic progenitors confirming the multipotency of isolated MSCs (Figure 1B).

### 3.2. Characterization of Human CIMVs-MSCs

CIMVs were successfully generated from primary human adipose MSCs (Figure 2). The morphology and size of the human CIMVs-MSCs were analyzed using scanning electron microscopy (SEM). CIMVs-MSCs had spherical structures and sizes ranging from 100 to 2600 nm with the majority (89.36%) having sizes between 100–1200 nm (Figure 2B).

Molecular composition of human CIMVs-MSCs was examined using proteome and xMap Luminex multiplex analysis (Figure 3 and Figure 4). Proteome analysis identified 373 proteins in human MSCs and 362 proteins in CIMVs-MSCs lysates. Interestingly, the majority (252 molecules) of proteins were similar between MSCs and CIMVs-MSCs, while 121 (32.4%) and 110 (30.4%) proteins were unique in MSC and CIMVs-MSCs, respectively (Figure 3A). The unique proteins of CIMVs-MSCs and MSCs are listed in Appendix A. 

The unique proteins in human MSCs were nuclear (21.3%), secreted (4.6%), lysosomal (1.9%), mitochondrial (17.6%), cytoplasmic/nuclear (23.1%), cytoskeleton (1.8%), cell membrane (9.3%), and cytoplasm (20.4%) location (Figure 3B). Peroxisome proteins were below the proteomics detection range in MSCs.

The unique proteins in CIMVs-MSCs included proteins associated with peroxisome (0.9%), lysosome (1.8%), mitochondria (6.5%), cytoplasm/nucleus (12%), cytoskeleton (20.4%), cell membrane (26%), and cytoplasm (32.4%) (Figure 3B). Nuclear and secreted proteins were below the proteomic detection range in CIMVs-MSCs.

A multiplex approach was used to characterize the molecular content of CIMVs-MSCs. Currently, there is little known about the CIMVs molecular content, while it could be suggested that it should retain, in part, the MSCs cytoplasm components, including intracellular cytokines. Due to these reasons, we sought to analyze the cytokine content of CIMVs as well as the parental MSCs. Multiple cytokines were similar between CIMVs-MSCs and parental MSCs. These included growth factors, cytokines, and chemokines (EGF, FGF-2, Eotaxin, TGF-α, G-CSF, Flt-3L, GM-CSF, Fractalkine, IFNα2, IFN-γ, GRO, IL-10, MCP-3, IL-12p40, MDC, IL-12p70, IL-15, sCD40L, IL-17A, IL-1RA, IL-1a, IL-9, IL-1b, IL-2, IL-4, IL-5, IL-6, IL-7, IL-8, IP-10, MCP-1, MIP_1a, MIP-1b, TNF-α, TNF-β and VEGF) (Table 1). Levels of IL-3 and IL-13 were below the detection range in MSCs and CIMVs-MSCs. Interestingly, levels of TGF- α, CCL7, sCD40L, IL-1b, and TNF-β were higher in MSCs as compared to CIMVs-MSCs.

### 3.3. Immunophenotype of Human CIMVs-MSCs

MSCs surface receptors play a role in cell to cell contact, immunomodulation, and activation of signaling in target cells [28]. Therefore, we sought to determine if CIMVs retain the surface receptors of MSCs. All parental MSCs (100%) expressed CD90, CD29, CD44, and CD73 (Figure 4) characteristics for the MSCs [27]. CIMVs-MSCs were positive for CD90, CD29, CD44, and CD73 (83%, 72%, 36%, and 66%, respectively) (Figure 4).

### 3.4. Transfer of Cell Surface Receptors to the Recipient Cell Membrane by CIMVs-MSCs

Microvesicles can transfer soluble factors as well as surface receptors by the fusion of cytoplasmic membranes [29,30]. Therefore, we sought to determine whether CIMVs-MSCs could transfer the surface receptors to the recipient HEK293FT cells. HEK293FT cells were pre-stained with DiO (Invitrogen, USA) and cultured for 24 h with DiD labeled CIMVs-MSCs (10 µg/mL) (Invitrogen, USA). Expression of CD90 was selected to demonstrate receptor transfer, as it is specific for CIMVs-MSCs and absent on HEK293FT cells. Expression of CD90 was analyzed using the laser scanning confocal microscope Zeiss LSM 780 (Carl Zeiss, Oberkochen, Germany) and flow cytometry BD FACS Aria III (BD Bioscience, San Jose, CA, USA). Donor MSCs demonstrated homogenous staining with DiD membrane dye (red fluorescence) and anti-CD90 antibody (blue fluorescence) (Figure 5A–C). We found that CIMVs-MSCs and HEK293FT membranes became fused and CD90 surface receptor was transferred to HEK293FT (Figure 5D–G). Regions of DiD and partial staining with anti-CD90 antibody staining were found in the cytoplasmic membrane of HEK293FT treated with CIMVs-MSCs (Figure 5D–G). Recipient cells (HEK293FT) treated with CIMVs acquired CD90 positive phenotype due to the internalization of CIMVs-MSCs membrane into the cytoplasmic membrane of the recipient cells. We determined that 99.14% of HEK293FT recipient cells acquired CD90+ immunophenotype (Figure 5H,I).

### 3.5. CIMVs-MSCs Stimulated Angiogenesis In Vivo

Since the CIMVs-MSCs contain multiple growth factors (EGF, FGF-2, and VEGF), we postulated that CIMVs-MSCs could have angiogenic activity. We used an in vivo approach to demonstrate angiogenetic capacity of MSCs and CIMVs-MSCs. MSCs (1 × 10^6^ cells) and CIMVs-MSCs (50 µg) were stained with vital membrane dye DiO (Invitrogen, USA), mixed with Matrigel matrix (400 µL), and injected into rats subcutaneously. Eight days later, MSCs and CIMVs-MSCs containing Matrigel matrix plugs were collected from the subcutaneous space of the rats and fixed in 10% formalin (group 1) or frozen in liquid nitrogen (group 2). Formalin-fixed Matrigel matrix plugs were stained with the hematoxylin-eosin kit (BioVitrum, Saint-Petersburg, Russia) (Figure 6A–C). Frozen Matrigel matrix plugs were cut using HM560 Cryo-Star microtome (ThermoScientific, Waltham, MA, USA) and stained with DAPI (D1306, Invitrogen, USA) (Figure 6D–F).

MSCs and CIMVs-MSCs were detected in the Matrigel matrix implants eight days after subcutaneous injection (Figure 6D–F). Moreover, newly developed blood capillaries were observed in Matrigel matrix containing MSCs and CIMVs-MSCs (Figure 6A–C). We found that the number of the newly developed blood vessels in control Matrigel matrix (without MSCs or CIMVs) was 0.67 ± 0.15 cap/mm^2^ (Figure 6A,D,G). In Matrigel matrix containing MSCs, the number of newly developed blood vessels was 11.3-fold higher (7.55 ± 0.46 cap/mm^2^, *p* < 0.01) than that in control (Figure 6B,E,G). Similar to MSCs, the number of the new capillaries in Matrigel matrix containing CIMVs-MSCs was increased (5.7-fold higher (3.84 ± 0.16 cap/mm^2^, *p* < 0.01)) than that in control (Figure 6C,F,G). We suggest that the human CIMVs-MSCs retain the angiogenic activity of the parental MSCs.

## 4. Discussion

CIMVs could be produced in large quantities while retaining the size of natural EVs [20,22]. However, our understanding of the biologic activity and therapeutic potential of MSCs-derived CIMVs remains limited.

Here, for the first time, we have shown that the size of the majority of human CIMVs-MSCs ranges between 100 and 1200 nm (89.36%), which is similar to that of EVs. Furthermore, the proteome content of human MSCs and CIMVs-MSCs appears to be similar. Analysis of the CIMVs-MSCs content revealed an increase of proteins linked to cytoskeleton, peroxisomes, cell membrane, and cytoplasm. In contrast, mitochondria and cytoplasm/nucleus proteins were decreased, while nucleus and secreted proteins were significantly depleted as compared to MSCs. We believe that the cytoskeleton proteins and membrane proteins enrichment of CIMVs-MSCs is due to the mechanism of their outward release from the cell surface. Similar data was demonstrated by Kim and colleagues [31]. We suggest that the enrichment of peroxisomes/cytoplasm proteins and depletion of mitochondria/cytoplasm/nucleus proteins could be due to the deep intracellular localization of these organelles.

MSCs activate cell proliferation, migration, and angiogenesis in vivo by direct contact and paracrine mechanisms including secretion of growth factors, cytokines, and chemokines [32]. Therefore, we sought to analyze the cytokine content and immune phenotype of human CIMVs-MSCs. We, for the first, time demonstrated that CIMVs-MSCs have a molecular content similar to that in parental MSCs (Table 1). Interestingly, CIMVs-MSCs had significantly lower levels of TGF-α (<0.02), MCP-3/CCL7 (<0.05), sCD40L (<0.05), IL-1b (<0.02), and TNF-β (<0.01). There is limited data on cytokine content of MSCs available. Several cytokines were detected by Mussano and colleagues in a MSCs culture medium. These included IL-2, IL-6, IL-8, IL10, IL-12, G-CSF, INF-γ, TNF-α, MCP-1 (CCL-2), IP-10, PDGF, bFGF, and VEGF [33]. The authors reported that MSCs produced high levels of IL-6, IL-8, MCP-1 (CCL-2), and VEGF [33]. Schinkothe and colleagues reported that human MSCs produced high levels of G-CSF, IL-12p40, IL-17, CCL2, CCL3, and CCL4 [34]. These data corroborate our results, where we have detected FGF2/bFGF, G-CSF, IFN-γ, IL-10, IL-12p40, IL-17A, IL-2, IL-6, IL-8, IP-10, MCP-1/CCL2, MIP_1a/CCL3, MIP-1b/CCL4, TNF-α, and VEGF in human MSCs and CIMVs-MSCs. The presence of several other cytokines in human MSCs and CIMVs-MSCs were demonstrated in our study, these include EGF, Eotaxin/CCL11, TGF-α, Flt-3L, GM-CSF, Fractalkine/CX3CL, IFNα2, GRO, MCP-3/CCL7, MDC/CCL22, IL-12p70, IL-15, sCD40L, IL-1RA, IL-1a, IL-9, IL-1b, IL-4, IL-5, IL-7, and TNF-β.

We found that CIMVs-MSCs have the surface receptors similar to that of the parental human MSCs: CD90^+^ (83%), CD29^+^ (72%), CD44^+^ (36%), CD73^+^ (66%). Our data corroborate results published by Pick and colleagues, where cell surface receptors were observed in the CIMVs membranes and their functionality was shown [20]. Kim and colleagues reported that the surface receptors were found to be similar between MSC-derived microvesicles and MSCs expressing CD13, CD29, CD44, CD73, CD105, CD10, and CD90 [31].

We have demonstrated that CIMVs-MSCs transfer membrane receptors to the target cells. It is known that MSCs surface cell adhesion molecules and signaling receptors play an important role in MSCs biology and maintain the stem like phenotype [35]. The surface receptor transfer by EVs could be the mechanism of mimicry and reprogramming of target cells. Similar data was published by Ratajczak and colleagues, where stem cell-derived microvesicles reprogram target cells by delivering their content including mRNA [36].

MSCs and CIMVs-MSC contained growth factors, cytokines, and chemokines, suggesting similar biological activity. To support this assumption, we have demonstrated that human MSCs and CIMVs-MSCs share the angiogenic activity. We have found that human MSCs and CIMVs-MSCs stimulate the sprouting of new blood vessels in vivo. Our data corroborate results published by Gangadaran et al. where MSCs-derived EVs increased cellular migration, proliferation, endothelial tube formation in vitro, and enhanced angiogenesis in ischemic limb in vivo [8]. In addition, Lopatina et al. reported that MSCs-derived EVs could induce the formation of vessel-like structures in vitro and in vivo after the subcutaneous injection in mixture with Matrigel Matrix and human microvascular endothelial cells [37]. We believe that the angiogenic capacity of CIMVs-MSCs depends on growth factors present in their content. Human CIMVs-MSCs demonstrated an angiogenic effect in vivo, although it was lower than that of MSCs parental cells. Due to the risks of MSCs therapy connected with undesirable differentiation [3,4] and transformation [4], the therapeutic use of cell-free therapeutic instrument based on CIMVs is only mechanistically feasible [12]. On the other hand, CIMVs-MSCs could be used to stimulate angiogenesis as they have molecular content and angiogenic activity similar to the parent MSCs. Therefore, CIMVs-MSCs could be used as a method for cell-free regenerative medicine.

## 5. Conclusions

We analyzed the molecular content, receptors expression, and angiogenic potential of human MSCs and CIMVs-MSCs. Human CIMVs-MSCs have similar content, immunophenotype, and angiogenic activity to that of the parental MSCs. CIMVs-MSCs could transfer membrane receptors to the surface of target cells. Therefore, we believe that human CIMVs-MSCs could be developed for cell-free therapy of degenerative diseases.

## Figures and Tables

**Figure 1 cells-09-00095-f001:**
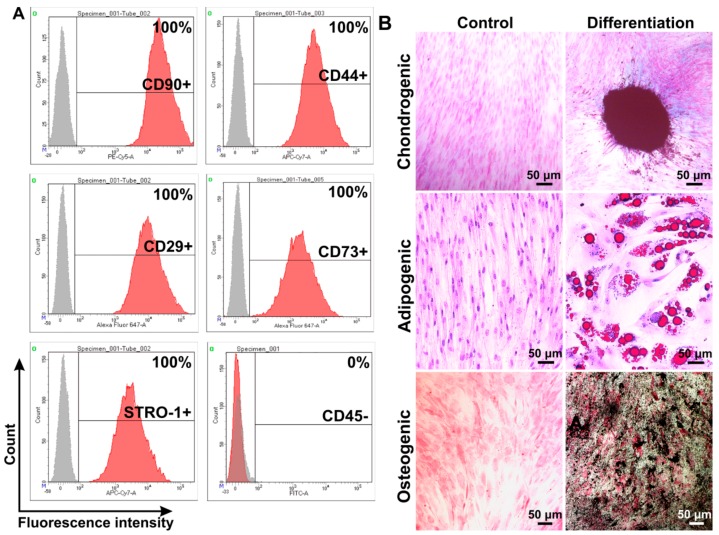
Phenotype analysis of human mesenchymal stem/stromal cells (MSCs) isolated from subcutaneous adipose tissue. Flow cytometry data (**A**). Histograms were generated using the FACSDiva7 software (BDBioscience, Version 7.0, San Jose, CA, USA). Gray—negative control; red—cells labeled with antibodies. Analysis of MSCs differentiation into: chondrogenic, adipogenic, and osteogenic progenitors (**B**). Differentiation was revealed by following stanings: chondrogenic differentiation—alcian blue, adipogenic—with oil red dye, osteogenic—silver nitrate staining. Images were captured using the ZEISS Axio Observer Z1 microscope (CarlZeiss, Oberkochen, Germany).

**Figure 2 cells-09-00095-f002:**
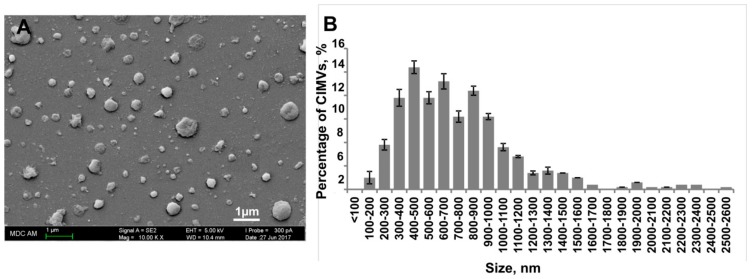
Analysis of the morphology and the size distribution of human cytochalasin B-induced membrane vesicles (CIMVs)-MSCs. Human CIMVs-MSCs were characterized using scanning electron microscopy (**A**). At least six electron microscope images were analyzed from three independent experiments to determine the size of human CIMVs-MSCs (**B**).

**Figure 3 cells-09-00095-f003:**
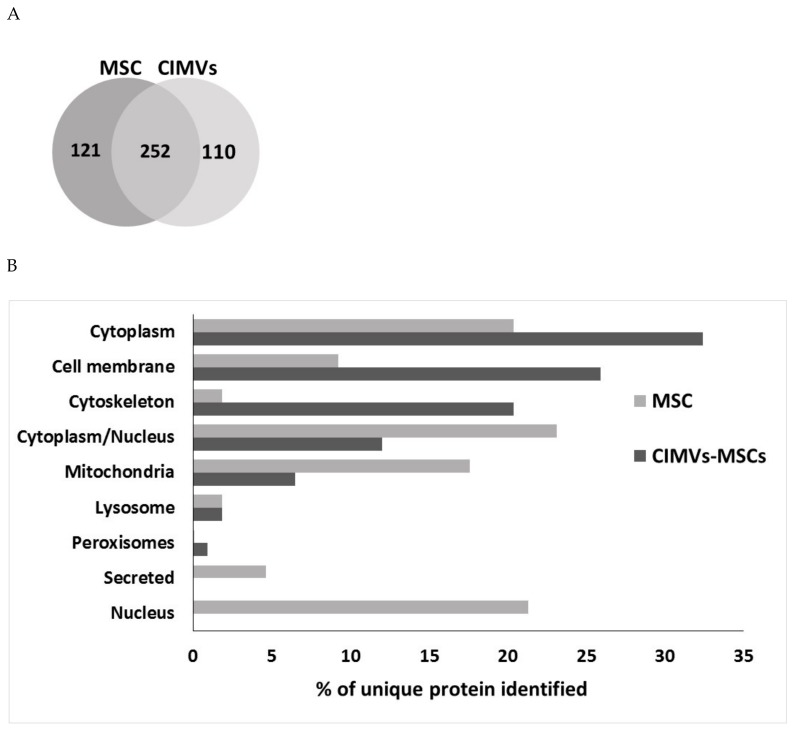
Proteome analysis of human MSCs and CIMVs-MSCs. Venn diagram of identified proteins MSCs and CIMVs-MSCs (**A**). Distribution of the identified proteins in organelles, % of unique identified proteins (**B**).

**Figure 4 cells-09-00095-f004:**
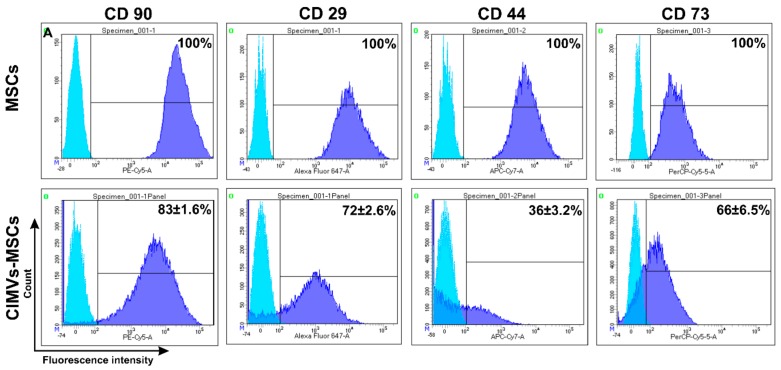
Immune phenotype of human MSCs and CIMVs-MSCs. MSCs and CIMVs-MSCs were stained with anti-CD90, anti-CD29, anti-CD44, and anti-CD73 monoclonal antibodies and analyzed using flow cytometer BD FACS Aria III (BD Bioscience, San Jose, CA, USA). Histograms were generated using the FACSDiva7 software (BDBioscience, Version 7.0, San Jose, CA, USA). Blue—isotype control; dark blue—MSCs or CIMVs-MSCs labeled with antibodies.

**Figure 5 cells-09-00095-f005:**
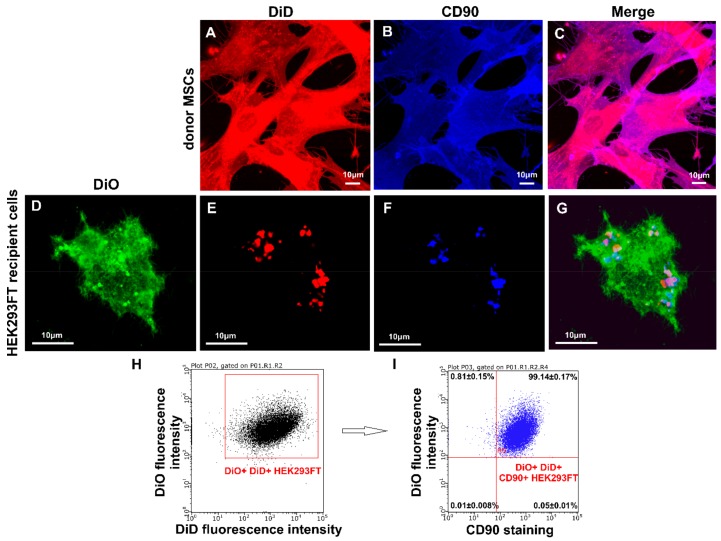
Analysis of CD90 transfer by CIMVs-MSCs to recipient HEK293FT cells. (**A–G**) Laser scanning confocal microscopy using Zeiss LSM 780 (Carl Zeiss, Oberkochen, Germany), (**H–I**) flow cytometry using BD FACS Aria III (BD Bioscience, San Jose, CA, USA). Green fluorescence—recipient HEK293 FT cell stained with DiO; red fluorescence—parental MSCs or CIMVs-MSCs stained with DiD, blue fluorescence—cells stained with anti-CD90 antibody. (**A–C**) MSCs stained with DiD and anti-CD90 antibody. (**D–G**) HEK293FT cells treated with CIMVs-MSCs (CIMVs-MSCs—red spots) and stained with DiO and anti-CD90 antibody.

**Figure 6 cells-09-00095-f006:**
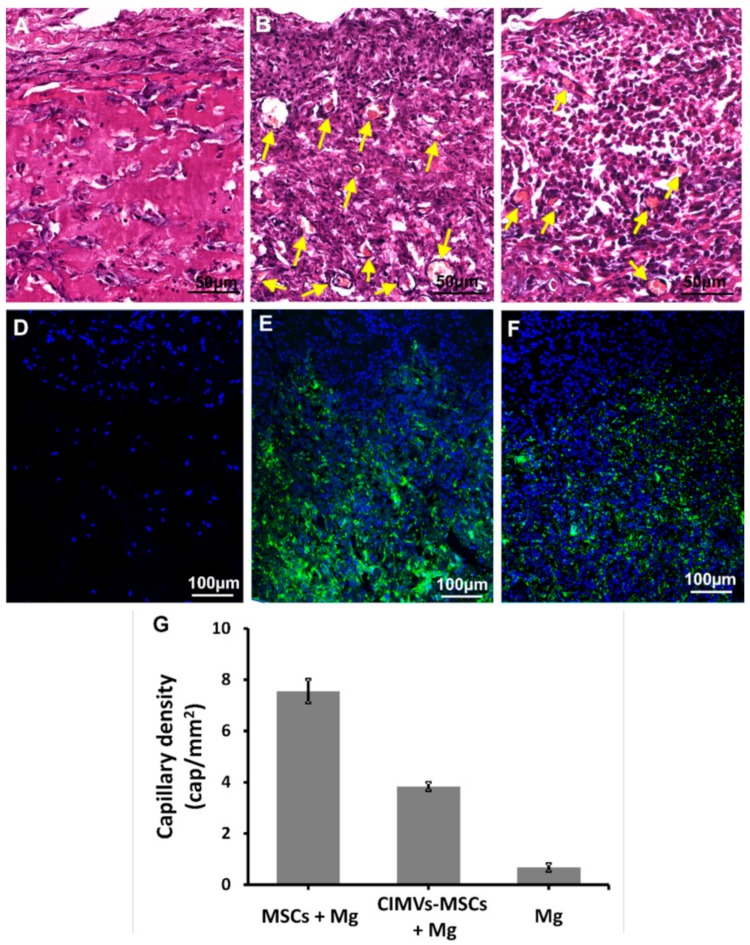
Analysis of angiogenic activity of MSCs and CIMVs-MSCs in vivo. Hematoxylin/eosin staining (**A–C**) and fluorescence micrographs (**D–F**) are shown after subcutaneous injections in rats (6 animals per experimental group) of MSCs (10^6^ cells) or CIMVs-MSCs (50 µg). (**A** and **D**) Negative control (subcutaneous injection of Matrigel matrix); (**B** and **E**) subcutaneously injection of MSCs; (**C** and **F**) subcutaneous injection of CIMVs-MSCs. MSCs and CIMVs-MSCs were stained with DiO (membrane dye) before the subcutaneous injection. Green fluorescence—DiO stained MSCs and CIMVs-MSCs; blue fluorescence—DAPI (stains DNA). Arrows mark the position of the sprouting blood capillaries. Counting of the total number of vessels was carried out using the AxioVision 4.8 program (CarlZeiss, Oberkochen, Germany). (**G**) Quantitation of the capillary density in Matrigel matrix plugs. The data represents mean ± SD. For statistical analysis, ten hematoxylin and eosin stained slides per animal were analyzed.

**Table 1 cells-09-00095-t001:** Cytokine analysis of MSCs and CIMVs-MSCs content.

Cytokine	MSCs (pg/mL)	CIMVs-MSCs (pg/mL)	*p* Value
EGF	141.3 ± 20.6	160.7 ± 67.7	<0.37
FGF-2	12114 ± 1433.9	10278.5 ± 1020.3	<0.14
Eotaxin/CCL11	48.6 ± 6.2	28.9 ± 22.4	<0.18
* TGF-α	3 ± 0.1	1.6 ± 0.3	<0.02
G-CSF	1327.4 ± 457.1	1144.3 ± 615	<0.38
Flt-3L	19.6 ± 21.4	19.3 ± 7.8	<0.49
GM-CSF	25.4 ± 6.5	15.6 ± 6	<0.13
Fractalkine/CX3CL	376.9 ± 28.4	548.7 ± 155.8	<0.07
IFNα2	162 ± 43.6	83.5 ± 61.7	<0.14
IFN-γ	7.5 ± 1.1	7.5 ± 3.8	<0.49
GRO	661.3 ± 234.1	278.6 ± 117.5	<0.06
IL-10	3.3 ± 1.1	3.4 ± 0.7	<0.47
* MCP-3/CCL7	60 ± 5	37.2 ± 9.3	<0.05
IL-12p40	31.4 ± 12.2	27.9 ± 13.4	<0.40
MDC/CCL22	20.7 ± 3.9	22.2 ± 6.4	<0.41
IL-12p70	7.4 ± 0.9	7 ± 2.9	<0.42
IL-15	18.7 ± 12.5	19 ± 5.8	<0.49
* sCD40L	9 ± 1.3	3.8 ± 2.2	<0.05
IL-17A	5.3 ± 0.9	2.2 ± 1.5	<0.06
IL-1RA	46 ± 35	40.6 ± 29	<0.44
IL-1a	13.2 ± 1.6	10.8 ± 4.8	<0.29
IL-9	136.3 ± 57.1	127.6 ± 68.3	<0.45
* IL-1b	8.7 ± 0.5	4.5 ± 1.1	<0.02
IL-2	9.9 ± 5	4 ± 1.9	<0.13
IL-4	18.8 ± 1.6	23.7 ± 13.1	<0.33
IL-5	3.8 ± 1.6	1.7 ± 0.2	<0.11
IL-6	1833.5 ± 235.4	1497.3 ± 513.8	<0.24
IL-7	21 ± 13.3	41.3 ± 16.3	<0.11
IL-8	1612.7 ± 487.2	721.3 ± 299.2	<0.08
IP-10	108.6 ± 44.9	268.1 ± 86.7	<0.07
MCP-1/CCL2	622.4 ± 238.9	1164.6 ± 505.1	<0.09
MIP_1a/CCL3	38.7 ± 2.5	28.5 ± 5.6	<0.07
MIP-1b/CCL4	82.4 ± 3.7	35.4 ± 29.8	<0.08
TNF-α	5.1 ± 1.3	2 ± 1.1	<0.06
* TNF-β	4.8 ± 0.3	2.8 ± 0.3	<0.01
VEGF	219.8 ± 90	215.4 ± 48.1	<0.48

* Cytokines, chemokines, growth factors which are enriched in MSCs compared to CIMVs-MSCs.

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
