# Peer review of "Angiogenic Activity of Cytochalasin B-Induced Membrane Vesicles of Human Mesenchymal Stem Cells"

_cells, 2019, doi:10.3390/cells9010095_

Round 1
Reviewer 1 Report
The manuscript from Gomzikova et al. is a study on the chemical-physical characteristics and the in vivo angiogenic properties of cytochalasin B-induced membrane vesicles derived from MSCs (CIMV-MSCs). The authors also investigate the ability of CIMV-MSCs to transfer membrane surface receptors from MSC-donor cells to recipient cells in in vitro experiments.
This manuscript is well written and experiments are adequate to the rationale of the paper. Although, some minor points should be addressed by the authors before publication:
There are a few grammar mistakes and typos in the text, please check the language. Since the advantages between cytochalasin induced- and non-induced membrane vesicles is already known from previous work, it would be more informative to report it in the Introduction where I could not find a sentence stating clearly why cytochalasin induced vesicles were chosen instead of traditional vesicles (i.e. the higher production of vesicles by MSCs with very similar properties to traditional ones). In this respect, as a suggestion, the first sentence in the Discussion section could be transferred to the Introduction. In the Materials and Methods sections (2.4 and 2.5) regarding the use of animals, the total number of animals used, the number of experimental groups and the number of animals used for each group is not explained. Please add this important information. In the results section (3.4) no mention to Fig. 5 panels A-C is present. Please add it following the sentence describing the experiment at line 252-253. Furthermore, in the legend of Fig 5 some information is lacking, what each fluorophore is and to which marker is associated with. In Fig. 6 is not possible to read the letters in panels A-C, maybe the authors could change the colour of the writings in the panels and of the arrows, black is not good. Furthermore, also in the legend of this figure some information is lacking about the fluorophores used in panels D-F and what they stain in the cells.
Author Response
Comment: There are a few grammar mistakes and typos in the text, please check the language.
Reply: As suggested by the Reviewer we have edited the grammar mistakes and typos.
Comment: Since the advantages between cytochalasin induced- and non-induced membrane vesicles is already known from previous work, it would be more informative to report it in the Introduction where I could not find a sentence stating clearly why cytochalasin induced vesicles were chosen instead of traditional vesicles (i.e. the higher production of vesicles by MSCs with very similar properties to traditional ones). In this respect, as a suggestion, the first sentence in the Discussion section could be transferred to the Introduction.
Reply: As suggested by the Reviewer we have transferred the mentioned sentence in the Introduction section (Lines 70-71).
Comment: In the Materials and Methods sections (2.4 and 2.5) regarding the use of animals, the total number of animals used, the number of experimental groups and the number of animals used for each group is not explained. Please add this important information.
Reply: We have clarified the number of experimental groups and the number of animals used for each group (Lines 195-197).
Comment: In the results section (3.4) no mention to Fig. 5 panels A-C is present. Please add it following the sentence describing the experiment at line 252-253.
Reply: We have added the description of the mentioned panels A-C to Fig. 5 (Lines 285-286).
Comment: Furthermore, in the legend of Fig 5 some information is lacking, what each fluorophore is and to which marker is associated with.
Reply: As suggested by the Reviewer we have added missing details in the Fig.5.
Comment: In Fig. 6 is not possible to read the letters in panels A-C, maybe the authors could change the colour of the writings in the panels and of the arrows, black is not good. Furthermore, also in the legend of this figure some information is lacking about the fluorophores used in panels D-F and what they stain in the cells.
Reply: As suggested by the Reviewer we have made the colors of the letters and arrows brighter and added missing details in the legend of Fig.6.
Reviewer 2 Report
The present manuscript describes the pro-angiogenic effect of so called membrane particles released by adipo-MSCs after stimulation with Cytochalasin-B, evaluating their size, content and effects.
The manuscript is of interest and the experiments well performed. There are a number of clarifications that need to be addressed.
The authors previously reported the encapsulation of chemotherapeutic drugs used to generate the membrane particles in the particles themselves and their toxic effects. Is cytochalasin present in the CIMV-MSCs? What is the difference between CIMVs and apoptotic bodies? The authors should add this clarification in the introduction, if known, or demonstrate it if still unknown.Author Response
Comment: The authors previously reported the encapsulation of chemotherapeutic drugs used to generate the membrane particles in the particles themselves and their toxic effects. Is cytochalasin present in the CIMV-MSCs?
Reply: This is about the study of Peng and colleagues (Peng et al. // Acs Applied Materials & Interfaces. 2015. V.7.). The authors applied an active loading of CIMVs with doxorubicin. In this study the membrane of CIMVs was reversibly permeated with digitonin.
In our study we did not permeate the membrane of CIMVs. We incubated MSCs in the presence of 10 µg/ml of Cytochalasin B. However as cytochalasin B binds with actin filaments, trace amounts of cytochalasin B may be retained within CIMVs associated with actin fragments. We previously evaluated HUVEC viability after addition of CIMVs (Supp.Information) (Gomzikova et al. // Oncotarget. 2017. V.8.). We found that there is not adverse effect on the recipient cells. CIMVs did not decrease vitality of the HUVEC.
Comment: What is the difference between CIMVs and apoptotic bodies? The authors should add this clarification in the introduction, if known, or demonstrate it if still unknown.
Reply: As suggested by the Reviewer we have added required information in the Introduction section (Lines 73-79). Previously it was shown that cytochalasin B does not cause the cells death, membrane vesicles possess surface charge similar to their parental cells and concluded that cytochalasin B-induced membrane vesicles are not apoptotic bodies (Peng et al. // Acs Applied Materials & Interfaces. 2015. V.7.).
Reviewer 3 Report
The main goal of the authors was to characterize CIMV-MSCs isolated from human adipose-derived MSCs in particular regarding their cytokine content and proteome. One of the major limitations of the study lies on the number of biological replicas. Given the well-know donor variation for MSCs it is paramount to validate the results in at least two more donors.
Line 19: mesenchymal stem cells should be mesenchymal stem/stromal cells. Moreover, in the case of the results described herein, the authors should refer to the cells as human adipose-derived mesenchymal stem/stromal cells.
Line 43: I suggest to focus on the “stem” cells used in the study which are MSCs and not other stem cell types such as ES or even iPS.
Line 45: reference 2 refers to foetal stem cells and not really MSCs and this could be misleading. I suggest to remove this reference.
Line 45: maldifferentiation if probably too strong based on the references (they only show that the cells differentiate into one but not another lineage and they did not test their differentiation potential extensively. Replace with limited differentiation potential sounds more appropriate.
Line 45: application is misspelled.
Line 46: provide examples that support the use of EVs in detriment of the producing cells. It is also necessary to provide more info regarding the different vesicles. For example, in line 47 the authors refer to EV but Line 50 already refer to micro vesicles.
Line 75: provide more details: size of the pieces, weight of the pieces used for digestion and final volume used for digestion. After 1h, the suspension was filtered or simply centrifuged?
Line 78: mention seeding density
Line 80: replace directions by lineages. Provide the composition of the different differentiation media and the seeding density and culture time for the different differentiation protocols.
Line 91: provide details regarding the culture conditions (seeding density for MCS, for how long the cells were grown before adding cytochalasin-B, were they trypsinised after the cytoD and if so for how long….)
Line 103: independent batches from independent donors or from the same donor?
Line 106: how many CIMVs-MSCs and how many MSCs?
Line 111: concentration of trypsin used.
Line 113: “by adding” should be replaced “by the addition”
Line 128: given the fact that FBS was used for the expansion of the cells the authors should mention if they used FBS-depleted from EVs or they should also consider the bovine database for their analysis.
Line 140: How much protein was used?
Line 190: mention in the legend the staining used for each figure.
Line 250: correct the units for DiD labeled CIMVs-MSCs.
Line 258: provide more details in the legend. For example, 5B and 5F represent what? 5E??? For the FACS plots show the controls (unstained cells, single labeled cells). Based on the figure it does not seem that the cells have homogeneous distribution of CD90.
Line 282: quantification of blood vessels should within the plug should be done with CD31 antibody or another endothelial specific marker. Explain why immune-competent rats were used given the fact that you transplanted human MSCs…
Line 315: Re-write the sentence (there’s a “.” in the middle and revise the structure of the sentence. There are other studies, not specifically focus in cytokines, for EV proteome analysis…
Round 2
Reviewer 2 Report
The authors adequately discussed my concerns.
Author Response
There are no comments. The manuscript was endorsed.
Reviewer 3 Report
Please, find below a few comments/suggestions:
Line 47: add an “a” after suggested.
Line 48: Replace “The” by An
Line 50: Rephrase: Suggestion: It was shown that MSC-derived EVs contain biologically active molecules, similar to the donor cell. In fact, several studies showed that EVs retain the therapeutic effect of the donor MSCs (6).
Line 53: after the (7) should have a full stop and the next sentence is already a referring to a different study.
Line 59: “this” should be “these”.
Line 60: the EV abbreviation has been used before. Check this throughout the manuscript.
Line 62: “EV are” should be replaced by “EVs comprise an” and “included” should be “including”
Line 65: Add a space after in between 100 and nm
Line 66: add “mostly” before contain.
Line 70: There are other strategies to enhance EV yield than just the one previously developed by the group. You should refer to those methods (e.g., micro carrier system, bioreactors, but also small molecules approaches that boosted EV secretion). These are just a few examples…As it is it seems cytochalasin B is the only method available.
Line 74: after was replace by “how different they were from aptoptotic bodies since it was known that citochalasin treatment could induce apoptosis” and delete the following sentence.
Line 75: remove “,” after demonstrated and “the cells death” should be “cell death”
Line 99: add space between 2 and mM.
Line 100: filtrated should be filtered.
Line 103: replace cultivation by culture and instead of the full stop proceed to “culture and the cells were maintained…”
Line 125: delete “action”
Line 297: was should be replaced by were.
The reply you provided for the CD90 should be included in the manuscript because it is rather important and can be missed. This is due to the presence of CD90 exclusively in the membrane of CIMVs-MSCs. Recipient cells (HEK293FT) treated with CIMVs acquired CD90 positive phenotype due to the internalization of CIMVs-MSCs membrane into the cytoplasmic membrane of the recipient cells. Therefore HEK293FT showed partial staining with CD90 and blue fluorescence.
Similarly, the explanation for the use of immune-competent rats should be included in the manuscript:
Immunocompetent animals were taken in the experiment due to the 1) low immunogenicity of MSCs (human and rat MSCs do not express class II MHC excluding them as antigen presenting cells to T CD4+ lymphocytes), 2) MSCs are able to reduce the immune response (Rossignol et al. // J Cell Mol Med. 2009. V.13.); 2) duration of the experiment was short and specific immunity on the introduced cells was not developed (Isakova et al. // PLoS One. 2014. V.9.).
This text below should also be included because it is highly relevant for the reader:
. We agree with the reviewer that we did not mention previous findings in the field of natural EV proteome analysis. This is due to the fact that CIMVs are different from natural EVs. It is known that the mechanism of EVs release provide sorting of bioactive molecules inside of EVs (Yuana et al. // Blood Reviews. 2013. V.27.) (Zhang et al. // Genomics Proteomics Bioinformatics. 2015. V.13.). The CIMVs production protocol does not provide the sorting of molecules inside of CIMVs. CIMVs enclose part of parental MSCs cell cytoplasm. Due to this reason in the present work we compared the cytokine content of CIMVs with parental MSCs.
Author Response
We have taken into account all the comments of the Reviewer and edited the manuscript.
Grammar mistakes and typos: Lines: 47, 48, 50, 53, 59-60, 62, 65, 66, 74, 78-79, 80, 103, 104, 107, 129, 309.
Required information has been added - Lines 70-75, 199-201, 262-264, 303-304, 309-312.